# Environmental Justice in the Context of Access to Urban Green Spaces for Refugee Children

**Siqi Chen** [1,*] and **Martin Knöll** [2,*]

1 Department of Architecture, School of Urban Construction & Safety Engineering, Shanghai Institute of Technology, No. 100 Haiquan Road, Fengxian District, Shanghai 201418, China
2 Urban Design and Planning (UDP), Department of Architecture, Technical University of Darmstadt, El-Lissitzky-Str. 1, 64287 Darmstadt, Germany
* Correspondence: siqi.chen@sit.edu.cn (S.C.); knoell@stadt.tu-darmstadt.de (M.K.);
  Tel.: +86-177-170-288-47 (S.C.)

**Abstract:** Accessible and high-quality urban green space (UGS) can provide significant benefits to refugee children for their development, health, and well-being. However, few studies have examined the actual accessibility of UGS from refugee children's perspectives (i.e., with restricted walking radius, particular vulnerability towards barriers such as traffic infrastructures and disconnected road forms) and related them with other environmental or social burdens under the context of environmental justice. It is necessary to explore related evidence and investigate the underlying causes since refugee facilities are primarily located in areas with restricted social and environmental resources strongly related to attributes of environmental justice. This paper investigated (1) availability, accessibility, and attractiveness of UGS in 30 refugee accommodation locations in Berlin using GIS and Space Syntax, (2) environmental burdens using the Berlin Atlas of Environmental Justice, and (3) neighbourhood characteristics. Findings indicate that 63% of refugee accommodations have availability of green space that is above average official standards, but from refugee children's perspectives, 60% of the locations have limited access to UGS, lower attractive green spaces, and most locations face multi-environmental burdens. Currently, little guidance focuses on ensuring equal access to and the usability of UGS for specific socioeconomic and demographic groups, such as refugee children. Therefore, this paper has contributed empirical materials to begin such research and develop inclusive decision-making strategies in environmental and health policy to ensure the provision and high quality of UGS for refugee children who need it.

**Keywords:** migrants; urban green space; built environment; urban health; environmental equality; accessibility; environmental burden; meso-environment; children-oriented; green space supply

## 1. Introduction

### 1.1. The Importance of Urban Green Space for Refugee Children

Urban green space (UGS) refers to an area within urban environments designated for recreational or aesthetic purposes, typically in the form of grass, trees, or other vegetation set, usually human-designed [1,2]. The positive effects of UGS on the health of children are well-recognised, including enhanced mood and self-esteem [3], a buffer from daily stress [4], and lower levels of depression and anxiety [5]. Refugee children may benefit more from UGS in reducing stress and improving mental health since UGS represents affordable urban recourse [6,7]. Research has also indicated that activities in the UGS act as a protective element in this population's overall well-being and health, offering emotional and physical sustenance in their transit periods [1]. At present, limited research has been conducted on the relationship between UGS utilisation and refugee health status in Europe, especially in refugee children [8,9]. Previous research provides evidence that parents may feel more comfortable if their children play in UGS since formal recreation facilities (in host countries)

are formed differently from their countries of origin [9]. The similarity of natural green spaces globally also makes green spaces familiar playfields for refugee children in their host countries, despite coming from diverse cultural backgrounds and regions [10]. Moreover, UGS could support their social inclusion by functioning as spaces where they connect with others [11]. Refugee children enjoy describing the games that they play in UGS and notice details such as bird tracking and flourishing flowers [9,12]. They also enjoy creating new rules for sporting activities in UGS, instead of formal regulations in place at sports sites [13]. From policy perspectives, the importance of UGS as an environmental contributor for minority groups (e.g., refugee children) has been recognised, as the European Union green infrastructure strategy emphasised the advantages of UGS in combating social isolation and enhancing community cohesion [14]. Furthermore, providing 'universal access to safe inclusive and accessible, green and public spaces' was also considered in the Global United Nations Sustainable Development Goal 11: Sustainable cities and communities [15].

### 1.2. Unequal Access to Urban Green Space from Refugee Children's Perspectives

Despite the benefits of green space for refugee children's development, evidence across Germany shows that children from migrant or refugee backgrounds are impeded from an equal chance to live in green neighbourhoods due to discrimination in the housing market [16,17]. Also, those neighbourhoods with low educational attainment, average income, and high unemployment rates typically have limited access to UGS compared to those with opposite characteristics [18,19]. Additionally, the World Health Organization Regional Office in Europe [20] reported that neighbourhoods with a high proportion of immigrant and ethnic minority populations tend to have less access to well-maintained green spaces. In the research scope of Berlin, refugees and immigrants tend to have limited access to UGS since they are settled in higher-density neighbourhoods [8,21].

Perceived safety from refugee parents is another critical factor influencing refugee families' access to UGS; migrant mothers settled in the UK expressed concerns about traffic problems when accessing UGS with their children [10]. Recent research on refugee children in Berlin indicated that refugee parents have particularly salient safety concerns (about green space) for their children's play and would rather keep them indoors [9]. Berlin's refugee accommodation staff also add new insights into how refugee families perceive danger in their immediate surroundings, and safety considerations by their parents are of particular significance. In other words, the locations where refugee children engage in play activities outdoors are partly under the supervision of or decided by their parents [22].

Additionally, differences in access to UGS across socioeconomic groups primarily depend on specific geographical locations [23], and the equitable distribution of UGS is influenced by urban planning and housing policies [24]. Until now, there has been little guidance on guaranteeing the accessibility and usability of UGS for specific socioeconomic and demographic groups, but a few cities have made efforts. For instance, Berlin seeks to establish the principle of environmental justice in its urban planning; that means avoiding the concentration of various environmental and social issues in particular neighbourhood areas [25], which will be discussed in the following subsection.

### 1.3. Access to Urban Green Space for Refugee Children in the Context of Environmental Justice

Quality of life and many aspects of the environment vary widely in different Berlin districts, particularly in the inner city. There are concentrations of health burdens, such as high population density (social burden) or insufficient green spaces, traffic noise, air pollutants, and heat islands (environmental burdens) [26]. In response to the multiple abovementioned burdens, the Berlin Environmental Justice system was built. The Berlin Environmental Justice system includes the analysis of five core indicators (noise burden, air pollution, thermal burden, green space supply, and social disadvantage) and two additional social indicators (residential status and degree of affecting inhabitants) from the interdepartmental environmental justice atlas, which aims to identify the districts subject to multiple burdens in Berlin. It serves as the basis for integrated strategies and measures

at the interface of environmental health and urban development [26]. The concept of environmental justice was first raised in 2019, and the State of Berlin became Germany's first metropolitan area to implement the concept. The health-oriented Berlin environmental justice approach is planning to become a facet of social justice, discussing the environmental development potentiality of disadvantaged areas, strengthening the social space-oriented administrative action in Berlin, and eventually, laying the basis for a new direction in environmental policy [25].

As one of the five core indicators for environmental justice in Berlin, the availability and accessibility to UGS have not yet been discussed sufficiently. Berlin's Environmental Atlas [27] follows the goal of providing at least 6 m$^2$ of UGS per inhabitant in densely populated urban areas. The identified obstacles are railway lines, large bodies of water, and motorways. However, the pure application of green space per person, accessibility, and threshold values does not offer a comprehensive evaluation of green space provision for an entire urban area [28], and it fails to reveal the distributions of green space across various segments of the population [21]. It is essential to evaluate and track particular demographic groups from diverse socioeconomic backgrounds in terms of their accessibility to green space [23]. It is more crucial to consider that the mere presence of green space alone does not ensure its utility since access to green space is contingent not only on its geographic proximity or accessibility (i.e., the existence of space within a reasonable distance from the investigated locations) but also on its quality (i.e., the existence and standards of facilities and amenities) [29,30]. When evaluating socioeconomic inequalities in the distribution of green space, the abovementioned two aspects should be considered, as well as the application of objective assessments and methods [29].

*1.4. Objective of This Study*

As Berlin, like many European capitals, grew extensively in terms of population and land over the last decade, refugees sought safety and shelter in the city, and so the number of temporary accommodations increased. The distribution and accessibility of UGS may not be equal across these vulnerable social population groups, and the literature remains largely silent on the green space supply from refugee children's perspective. On this basis, this paper discusses the possible injustice of UGS provision among refugee children through thirty refugee accommodations in Berlin. The green space supply for refugee children was analysed from distributive dimensions (availability, accessibility, and attractiveness), and the findings were then related to procedural and interactional elements of Berlin's environmental justice. Nevertheless, the primary objectives are:

- To indicate possible unequal access to UGS of refugee children and all inhabitants by analysing the perceived distribution of UGS within thirty study sites;
- To identify and discuss other environmental burdens faced by refugee accommodation locations according to Berlin's environmental justice framework;
- To provide guidance on managerial considerations for the planning and implementation of UGS, with specific reference to the environmental justice of Berlin.

## 2. Materials and Methods

We employ a 3-step mixed-method approach and begin with methods updated from the official Environmental Atlas Berlin benchmark [27] to evaluate access to UGS from refugee children's perspective.

*2.1. Study Site*

Berlin is located in the eastern area of Germany, and its administrative boundaries hold more than 89,000 ha. In order to better plan, study, and foresee future socio-demographic development, Berlin works with subdivisions of "every day environments" (Lebensweltlich Orientierte Räume, German acronym: LOR), which are organised in 3 different scales corresponding to levels of regional, city, and community planning. At the most small-scale level, there are 542 neighbourhoods, or planning living areas (PLAs), which reflect homogenous

building typology, and vary from 21 to 16,000 inhabitants [31]. By 2022, 3,755,251 inhabitants lived in the metropolitan area of Berlin [32]. A total of 40% of 85,000 registered asylum seekers are estimated to be children under 18 years old [33]. Four primary accommodation categories are available to asylum seekers in Berlin before securing permanent residency (Figure 1). In response to the rising numbers of protection seekers from Afghanistan and Ukraine, the Berlin Senate extended the duration of *emergency accommodations* (German: Notunterkunft) until March 2023 [34,35]. For instance, the capacity of the former airport Tegel was extended to 1900 (Terminals A and B [36,37]) plus 3200 places (Terminal C [38]) after comprehensive reconstruction. Asylum seekers are generally mandated to reside in an *initial reception* (German: Erstaufnahmeeinrichtung) after submitting their applications. They could stay here for a maximum of 24 months [39,40]. In Germany, Federal States such as Berlin (it is a city and state at the same time) are responsible for establishing and maintaining *initial receptions* [41], which are typically affiliated with a branch organisation of the Federal Office for Migration and Refugees (BAMF). *Emergency accommodations* are usually operated together with *initial receptions* as different refugee accommodations; in other states, they are also called *arrive centers* [42]. *Tempohomes* (constructed residential containers) are one of the specific refugee accommodations that exist only in Berlin, and they are meant to serve as a temporary solution until more stable living arrangements become available [43]. Once asylum applications proceed, asylum seekers will be accommodated in *community accommodations* (German: Gemeinschaftsunterkünften) until permanent accommodation is available [44], which typically happens following the initial reception period. They are required to stay in the assigned district following the 'geographical restriction' for the entire duration and appeal proceedings (see freedom of movement) [39,45].

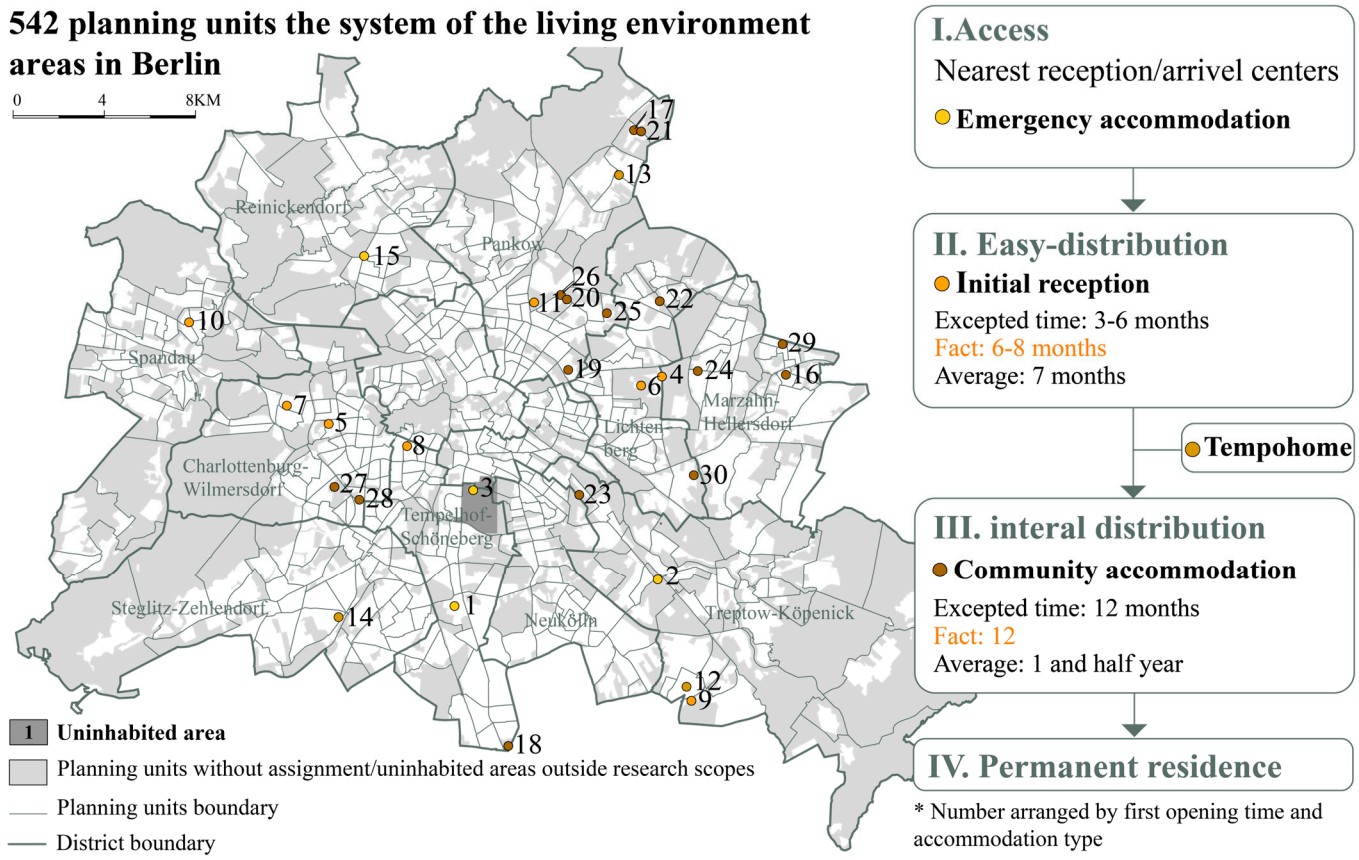

**Figure 1.** Investigated refugee accommodation types, locations, and located PLA in Berlin.

The present study endeavours to explore neighbourhoods surrounding three emergency accommodations, eight initial receptions, three Tempohomes, and sixteen community accommodations. Even though 1/5 of the study sites were closed by the time of data sum-

mary (February 2024, Table 1), it is still necessary to investigate them at comparable levels since the necessities of document-related evidence and the closing accommodations show stakeholders' choice proportions and social contexts (details in Supplementary Table S1, such as data collection period of each location).

**Table 1.** Overview of investigated refugee accommodations and their neighbourhoods.

| Site | Period | Districts | Children's Number [1] | Countries of Origin [1] | Accommodation Type |
|------|--------|-----------|------------------------|--------------------------|---------------------|
| 1 | 2015–now [2] | Tempelhof-Schöneberg | | Non-specific | Emergency accommodation |
| 2 | 2015–2018 | Treptow-Köpenick | - | Middle East and North Africa | |
| 3 | 2015–now | Tempelhof-Schöneberg | | Non-specific | |
| 4 | 2012–now | Lichtenberg | 27 | Non-specific | Initial reception |
| 5 | 2013–2018 | Charlottenburg-Wilmersdorf | - | | |
| 6 | 2014–2019 | Lichtenberg | 33 | Asia and Africa | |
| 7 | 2015–now | Charlottenburg-Wilmersdorf | - | Syria, Iraq and Afghanistan | |
| 8 | | Tempelhof-Schöneberg | | | |
| 9 | | Treptow-Köpenick | 30 | Non-specific | |
| 10 | 2017–now | Spandau | - | | |
| 11 | 2019–now | Pankow | | | |
| 12 | 2016–now | Treptow-Köpenick | - | Non-specific | Tempohomes |
| 13 | 2016–2019 | Pankow | 20–30 | | |
| 14 | 2017–now | Steglitz-Zehlendorf | - | | |
| 15 | 2013–now | Reinickendorf | - | Middle East, Ukraine, and Russia | Community accommodation (for vulnerable groups) |
| 16 | 2013–now | Marzahn-Hellersdorf | - | Non-specific | Community accommodation |
| 17 | 2015–2020 | Pankow | 30 | | |
| 18 | 2015–2020 | Tempelhof-Schöneberg | 18 | | |
| 19 | 2015–now | Pankow | 70–80 | | |
| 20 | | Pankow | - | | |
| 21 | 2017–now | Pankow | | | |
| 22 | 2018–now | Lichtenberg | 150 (<18 Y) | | |
| 23 | | Neukölln | - | | |
| 24 | 2020–now | Marzahn-Hellersdorf | | | |
| 25 | | Pankow | 19 | | |
| 26 | 2021–now | Pankow | | | |
| 27 | 2022–now | Charlottenburg-Wilmersdorf | - | Middle East, North Africa and Eastern European | |
| 28 | | Charlottenburg-Wilmersdorf | | | |
| 29 | | Marzahn-Hellersdorf | | Non-specific | |
| 30 | | Lichtenberg | | | |

[1] By demographic data collection/interview phase, more details in Supplementary Table S1. [2] Until February 2024.

*2.2. Database*

The Environmental Justice of Berlin (EJB) from the Berlin Senate Department for Urban Mobility, Transport, Climate Action and the Environment [25] was chosen to describe the environmental and socioeconomic status of 30 study sites. Part of the findings in this paper develop an analysis and aggregation of available databases from EJB. The EJB approach assesses environmental justice by relying on the available data or existing analysis framework of Berlin (e.g., monitoring social urban development indicators [46]). Five core indicators are investigated in this step, including noise burden, air pollution, thermal burden, green space supply, and social disadvantage. The first three indicators mentioned above are related to health risk weighting, while the other two are unrelated. Neighbourhoods that experience exceptionally high environmental burdens (multi-burdened areas, twofold-burdened, and so on), which means that study sites with environmental and/or social disadvantages could be revealed by this step. In the next step, the Berlin Environmental Justice map will be completed with additional environmental and social information. One is local population density, which refers to the degrees to which inhabitants are affected by core environmental and social indicators. Another one is the rent index, which refers to the residential area's status. These identifications may also be utilised to prioritise the areas affected by multiple factors according to the urgency of action. Table 2 explains the reference period, information on interpretation (for the indicators and indices), and data sources for the abovementioned indicators applied in this research. Additionally, information regarding Tempelhof Airport (Site 3, German: Tempelhofer Feld) consists only of data regarding green space supply from refugee children's perspective (Section 2.3).

**Table 2.** Data characteristics of the reference period, calculation method, and data source.

| Level | Dimension | Data | Reference Period | Interpretation | Data Source |
|---|---|---|---|---|---|
| Core Indicators | Noise burden | Strategic Noise Maps [1] | 2021 | Traffic noise sources as total level addition at night (from 10 p.m. to 6 a.m.), which goes beyond the Environmental Noise Directive requirements | Senate Department for Urban Mobility, Transport, Climate Action, and the Environment |
| | Air pollution | $NO_2$ measuring stations; $PM_{2.5}$ modelled data | 2018–2019 | The almost 50 $NO_2$ measuring points were statistically interpolated on a 100-metre grid, considering the building structure [2] and the traffic volume with regression analysis. | Senate Department for Urban Mobility, Transport, Climate Action and the Environment |
| | Thermal burden | Climate Model Berlin; Physiological Equivalent Temperature; Air Temperature Map | 2015 | An evaluation of "summer heat stress" with a $10 \times 10 \text{ m}^2$ grid, assessing both during the day, at the time of the sun's highest point (2 p.m.), and at night (4 a.m.) | Senate Department for Urban Development and Housing |
| | Green space supply | Analysis of the urban availability of green space (VAG) | 2020 | Availability of Public, Near-residential Green Spaces 2020; refer to Table 3 | Senate Department for Urban Mobility, Transport, Climate Action and the Environment |
| | Social disadvantage | Social Urban Development Monitoring | 2021 | The following three index indicators form the basis for the calculation of the status and dynamics index (two-year development): 1. Unemployment [3]; 2. transfer payments of the non-unemployed [4]; 3. Child poverty [5]. | Senate Department for Urban Development, Building and Housing |

**Table 2.** *Cont.*

| Level | Dimension | Data | Reference Period | Interpretation | Data Source |
|---|---|---|---|---|---|
| Add | Residential status | Rent Index | 2021 | More than 66% of the affected residential addresses (planning areas with simple residential character) | Senate Department for Urban Development and Housing |
| | Degree of affecting inhabitants | Population density | 2021 | Areas with more than 10,000 residents/km$^2$ refer to the environmental and social stress factors with the number of affected persons | Statistical Office for Berlin-Brandenburg |
| Environmental Justice | | Refers to all core indicators and additional information | 2022 | Combination of all dimensions above | Refers to all core indicators and additional information |

[1] Includes recalculations of noise reductions due to the closure of Tegel Airport. [2] Floor Space Index and Site Occupancy Index. [3] According to SGB II (Social Code—Book II). [4] According to SGB II and XII. [5] According to SGB II of under 15-year-olds.

**Table 3.** Iterative approach of access to UGS from refugee children's perspective.

| Dimension | Calculation Approach | Calculation Method |
|---|---|---|
| Available green space | Green space within 1000 m of the target study site | Circle locating method based on roundness |
| Accessible green space | *Available green space* located less than 500 m and 500–1000 m perceived neighbourhood distances away from the target study site | Perceived neighbourhood distance and accessible spaces tool from the authors' previous research [20] |
| Attractive green space | *Accessible green space* on the road that has the top 20% global integration among all investigated road segments. | Space syntax, the global integration value |

*2.3. Methods*

2.3.1. Availability, Accessibility, and Attractiveness of UGS for Refugee Children

In line with Kronenberg [47] and his colleagues, the analysis of UGS (green space supply or access to UGS) for refugee children in this paper was developed and adapted from their perceptions using an iterative approach with a novel perceived calculation approach and method, as highlighted in Table 3. The present method considered quantitative calculation updated from qualitative evidence of the *availability, accessibility, and attractiveness* of UGS from refugee children's perceptions. *Available green space* refers to green space within 1000 m from the target study sites from geographical information system (GIS) of OpenStreetMap and Land Use Plan of Berlin. *Accessible green spaces* are available green spaces located less than 500 m (children's perceived neighbourhood distance) and 500–1000 m (parents' perceived neighbourhood distances) away from each target study site. The perceived distance is developed from the authors' previous refugee children and their parents perceived distance tool [8], considering their specific concerns of safety and the presence of accessible green space [9]. *Attractive green spaces* are accessible green spaces on the road that has the top 20% global integration (space syntax method) among all investigated road segments and with an area of not less than 0.5 ha (please refer to Table 4 below); higher global integration refers to better-connected road segments, and the qualitative evidence indicated that refugee children and their parents are likely to take well-connected roads instead of disconnected (abandoned) roads, also for safety reasons [9].

It is worth mentioning that the catchment area and scope for green spaces are smaller for refugee children compared to all habitants since refugee parents and children have a different, more anxious perception of safety [8,9]. The assessment process diagram according to green space supply for refugee children can be found in Figure 2, with *Site 7* as an example. More details on mathematic procedures are presented in Supplementary Table S3.

**Table 4.** Green space supply from all inhabitants' and refugee children's perspectives.

| Dimension | All inhabitants' Perspective | Refugee Children's Perspective |
|---|---|---|
| **Data source** | Land Use Plan of Berlin [1] | OpenStreetMap and Land Use Plan of Berlin |
| **Space feature** | Green spaces (GRIS)<br>Playgrounds (GRIS);<br>Green spaces as compensatory measures (e.g., park);<br>Areas maintained by Grün Berlin GmbH (e.g., garden);<br>Forest areas;<br>Private green/open space | Green spaces;<br>Playgrounds;<br>Parks;<br>Gardens;<br>Forest areas;<br>Private green/open space |
| **Distinction** | **Near-residential** | **Perceived neighbourhood distance for refugee children** |
| **Scope** | ≦500 m | |
| **Distinction** | **Near-development** | **Perceived neighbourhood distance for refugee parents** |
| **Scope** | 1000 m to 1500 m | 500 m to 1000 m |
| **Area size (ha)** | ≥0.5 | |
| **Calculative methods** | catchment area surrounding the residence (railway lines, large bodies of water, and motorways are excluded) | perceived neighbourhood using an iterative approach (only footways are included) |
| **Calculation degree** | The classification is based on the calculated standard value (SV) in Berlin of public green spaces of 6 m²/per person.<br>Green space supply "good" (+): x >= SV<br>Green space supply "medium" (±):50% SV < x < SV<br>Green space supply "poor" (−): x < 50% SV | The classification is based on average Attractive green space (AGS) numbers.<br>Green space supply "good" (+): x >= AGS<br>Green space supply "medium" (±): AGS < x < 2 AGS<br>Green space supply "poor" (−): x < AGS |
| **Area size (ha)** | ≥0.5 | ≥10 |

[1] German: Flächennutzungsplan Berlin (German acronym: FNP).

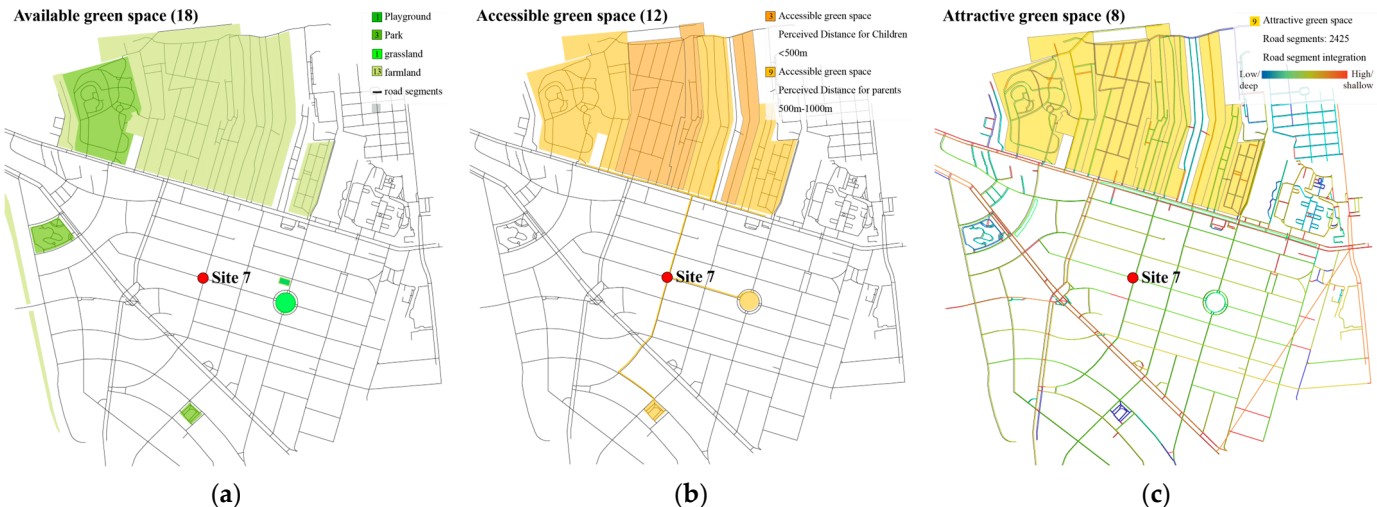

(**a**)    (**b**)    (**c**)

**Figure 2.** Assessment process according to green space supply of study Site 7: (**a**) available green space; (**b**) accessible green space; (**c**) attractive green space.

### 2.3.2. Green Space Supply for Refugee Children

Table 4 shows the comparable method of the official green space supply from all inhabitants' perspectives and the authors' modified method from refugee children's perspectives. Due to the temporary period of refugee accommodation, the authors also applied an additional database from OpenStreetMap [48] as a supplement to official green space availability

of Public, Near-residential Green spaces from Environmental Atlas Berlin [27]. The detailed space feature coding applied is listed in Supplementary Table S2. Berlin applies a guideline of 500 m (defined as near-residential) to a UGS of at least 0.5 ha. For green spaces bisected by streets, the resulting segments are considered if one is more significant than 0.5 ha. The scope of 1 to 1.5 km is defined as near-development; residents should be able to access more prominent green space areas of at least 10 ha. Additionally, each resident in Berlin should be able to access at least 6 m² (small) or 7 m² (large) green areas [27] as the calculated standard value (SV). This calculated value is updated to average Attractive green space (AGS, as mentioned in Table 4) numbers from refugee children's perspectives.

2.3.3. Environmental Justice in the Context of Access to UGS for Refugee Children

Figure 3 represents a comparable analysis of the official and updated environmental justice approaches from the perspective of refugee children. The EJB evaluation from 30 study sites will produce an overview of the environmental and social status of refugee accommodations. Both analysis approaches will add novel evidence from refugee children's perspective. This diagram also illustrates the methodological approach of environmental justice in the context of access to urban green spaces for refugee children with a three-step mixed-method approach. *Step 1*, the green space supply indicator, was redefined from the perspective of refugee children with an iterative procedure to scope UGS from availability, accessibility, and attractiveness values. In *Step 2*, we analyse thirty study sites under the official environmental justice framework of Berlin to evaluate the comprehensive social and environmental burdens of selected refugee accommodation locations. Results are compared with official green space supply distribution, and analysis identifies whether dissimilarities of UGS provision exist. In *Step 3*, three representative study sites are compared with their actual neighbourhood situations. The clear advantage of employing scales—site-specific neighbourhoods (1000 m from targeted refugee accommodation), planning areas, and city scale—is that we are capable of comparing city-scale threshold values with the unique requirements of the local population within a specific settlement site [21].

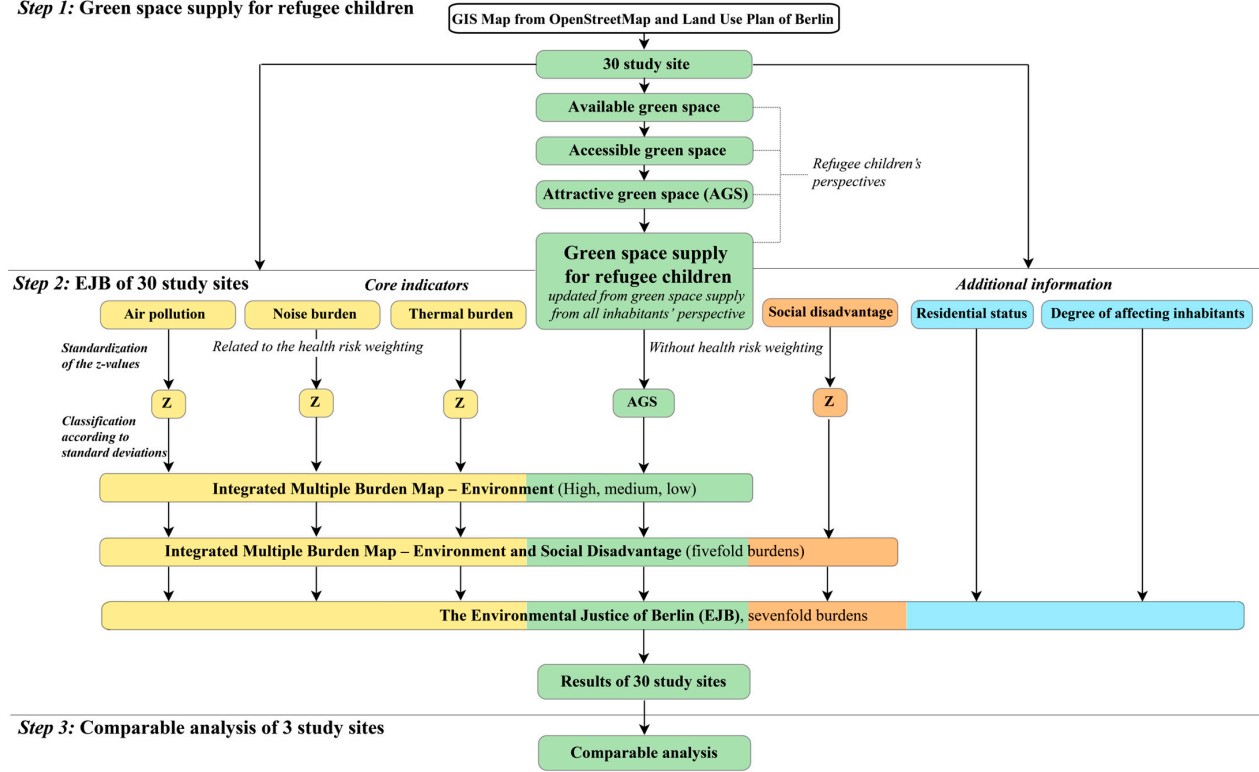

**Figure 3.** Methodological approach of environmental justice in the context of access to urban green spaces for refugee children.

## 3. Results and Comparison

### 3.1. Comparison Findings from Green Space Supply

Figure 4 illustrates the comparable findings of green space supply from the perspectives of all inhabitants and refugee children. *Sites 4, 6, 7, 17, and 23* have sufficient access to UGS (status "good") from both perspectives, and *Sites 5, 8, 10, and 28* have insufficient access to UGS (status "poor") from all states. Since the current paper applies strict access standards to UGS from refugee children as perceived evaluations of UGS availability, accessibility, and attractiveness (Section 2.3, Tables 3 and 4), 60% of sites have lower access to UGS from refugee children's perspective. Moreover, from refugee children's perspectives, seven sites, *Sites 2, 9, 12, 14, 22, 25, and 30*, are reduced from "good accessibility" to "poor accessibility" and therefore require special attention since refugee children have lower potential opportunities for UGS compared to all other residents. The abovementioned sites are all distributed outside the city centre (out of the city rail circle line), and most are located in Lichtenberg and Treptow-Köpenick. Moreover, six sites (*Sites 1, 15, 19, 20, 26, and 27*) are reduced from "medium accessibility" to "poor accessibility" based on refugee children's perceptions. Furthermore, all sites that stay in the "medium" status (Figure 4a) from all inhabitants' perspectives are reduced to "low" status (Figure 4b) when it comes to refugee children's perspectives. From our investigations, the classification of the green space supply from both perspectives (all inhabitants and refugee children) shows an unbalanced distribution of access to UGS. In summary, Figure 4 illustrates the poor accessibility of UGS available for refugee children from their perspectives.

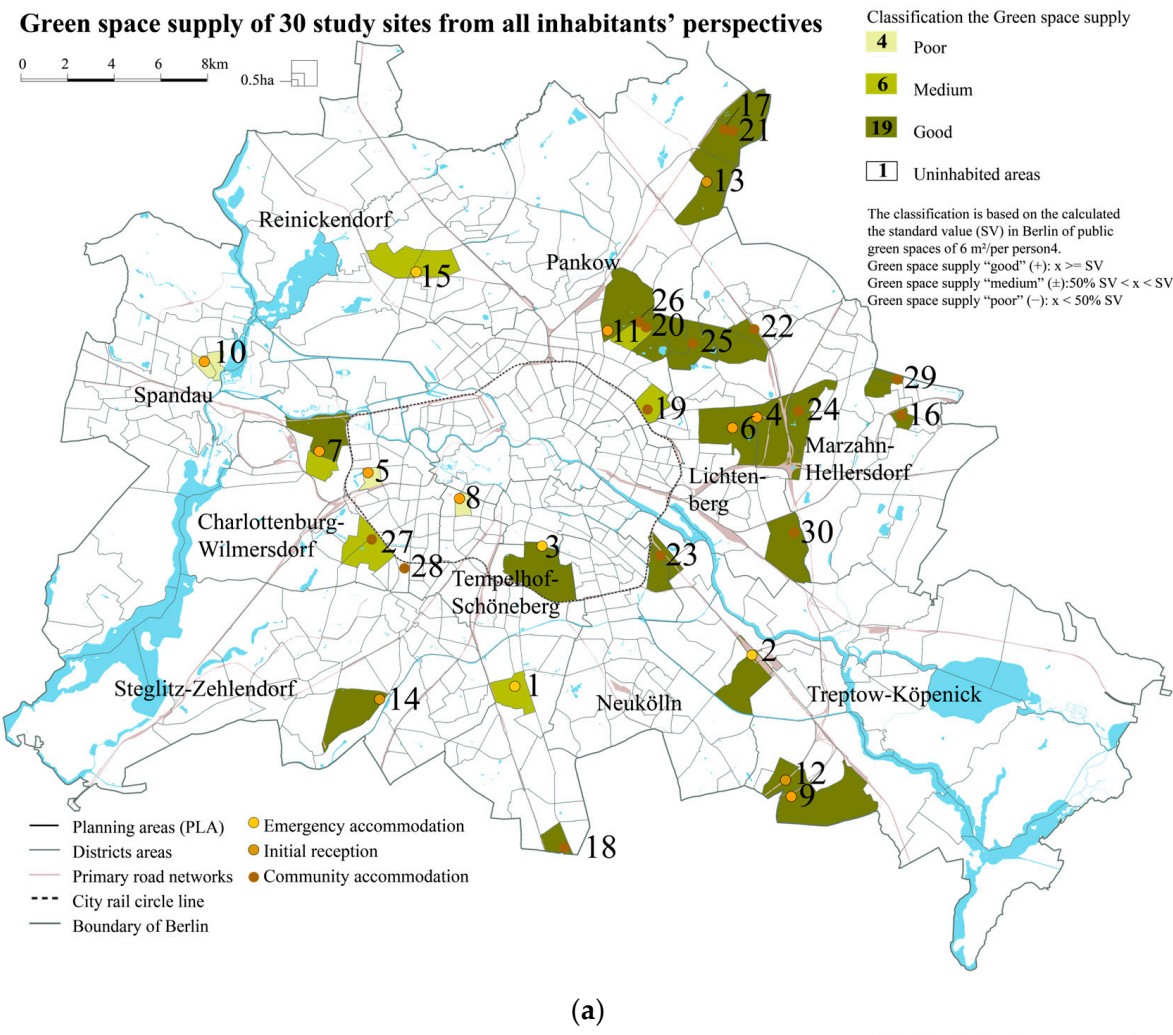

(**a**)

**Figure 4.** *Cont.*

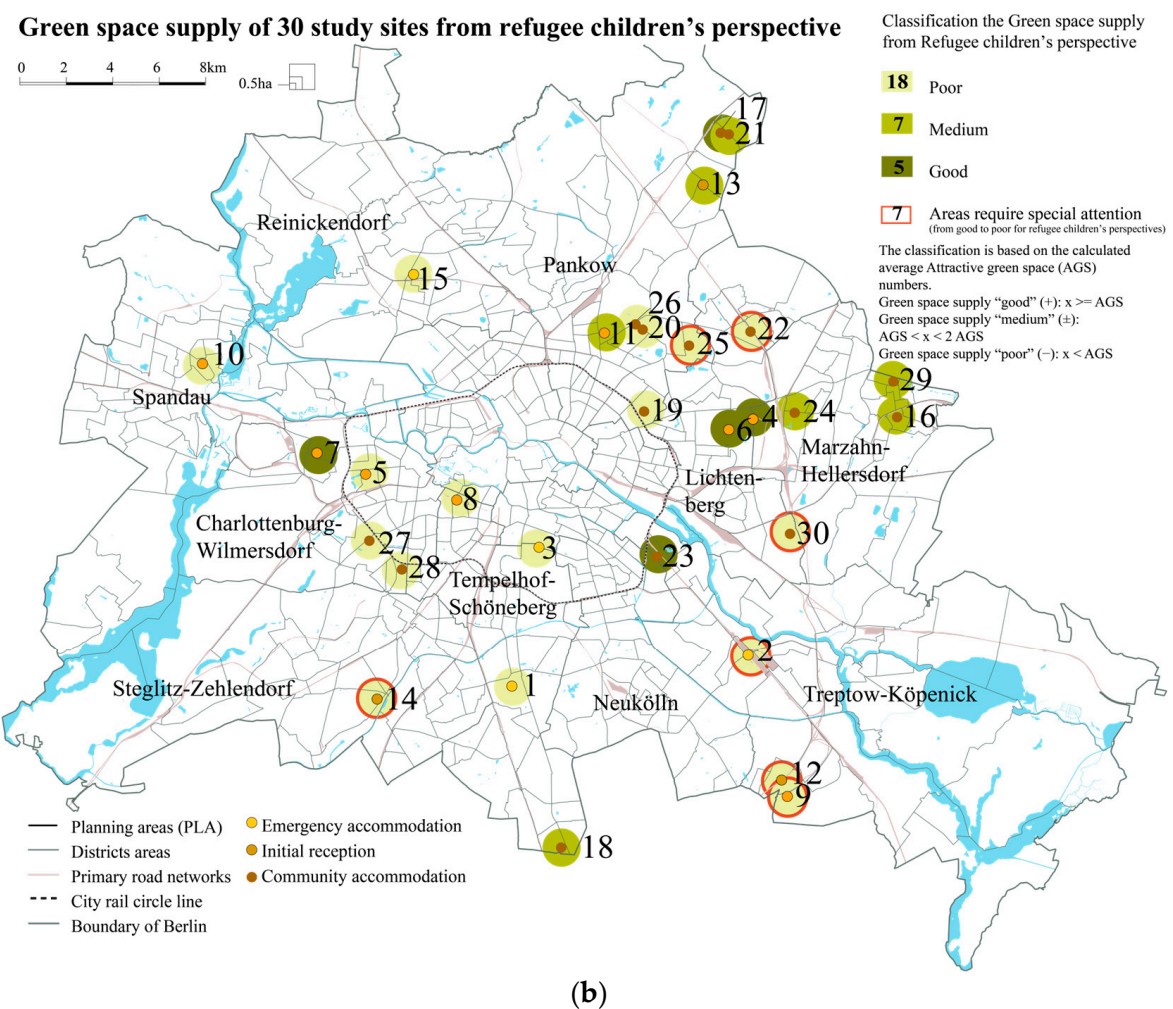

**Figure 4.** Green space supply of 30 study sites (**a**) from all inhabitants' perspectives; (**b**) from refugee children's perspectives.

### 3.2. Comparison of Findings from the Environmental Justice Map of Berlin

The data analysis of 542 PLAs is published in the Geoportal as series maps and compiled in a comprehensive set [49]. This study presents and describes statistical analyses of 30 site studies (Figure 5). It entails the analysis of the five-part core indicator and the multiple burden maps; the burdens indicate the site situation at the entire city level with quantitative and qualitative approaches. This analysis shows current social cohesion (neighbourhood management areas), the effectiveness of district planning, and budget calculations for selected social infrastructure facilities in value equalisation. Disadvantaged neighbourhoods, which means areas requiring special attention from Urban Development, Building and Housing and cross-departmental communities in Berlin [26], are also highlighted (with border boundaries) in this diagram.

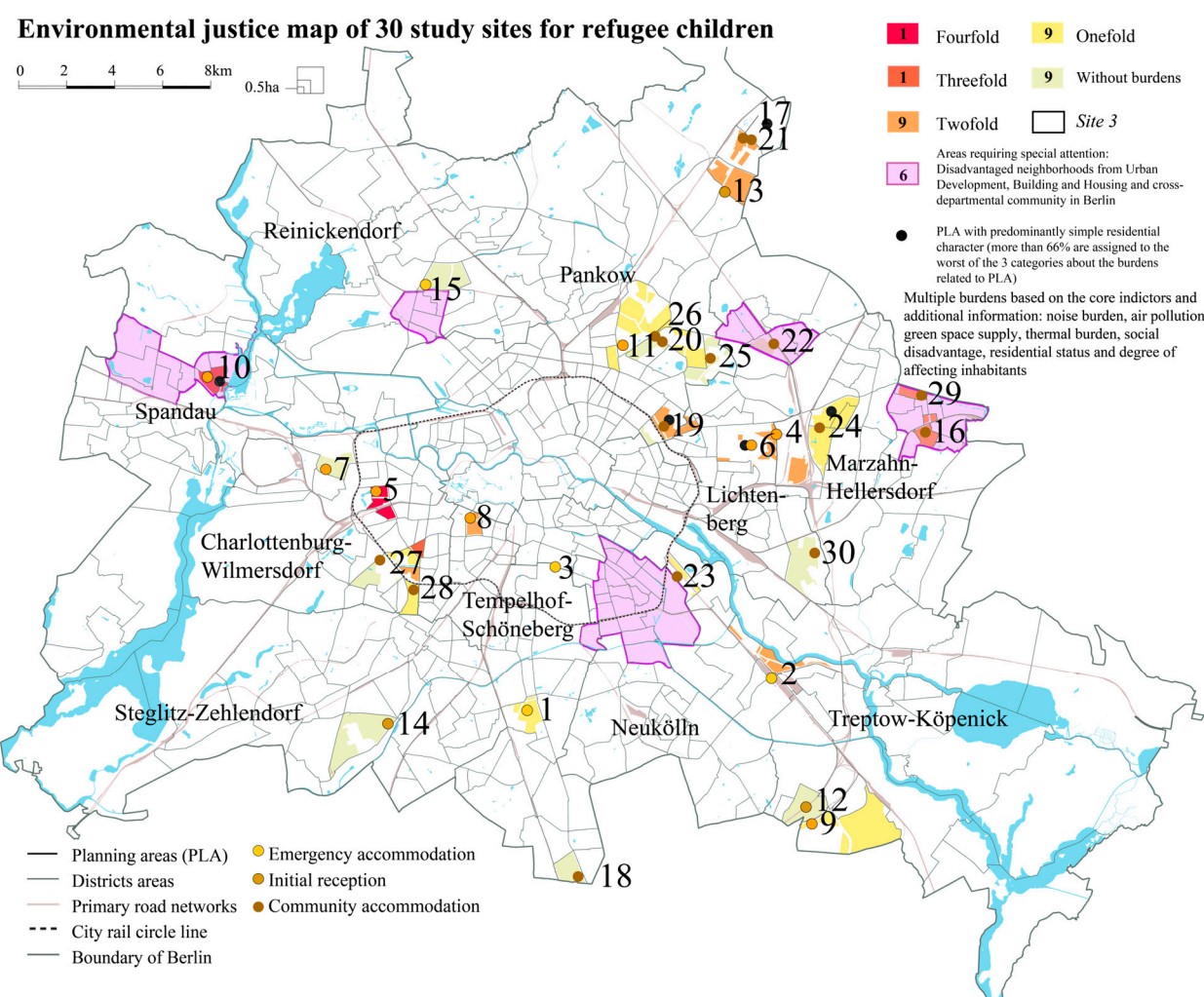

**Figure 5.** Environmental justice map of 30 study sites for refugee children.

Moreover, as mentioned before (Section 2.2), the former airport *Site 3* was only calculated with green space supply for refugee children and superimposed on this diagram as an "uninhabited area". The findings indicate that there are nine sites (*Sites 12, 14, 15, 18, 20, 25–27, and 30*) without environmental burdens; however, combined with findings from the last section, only *Site 18* transforms from good accessibility to medium accessibility when seen through refugee children's perspective, and would therefore still qualify as "without burdens" in EJB. If the calculation were performed from refugee children's perspective of UGS, the other eight sites would be calculated as a "onefold burden" since they have poor access to UGS in refugee children's views. Returning to the EJB diagram, 70% (21 sites) have at least one burden. Table 5 compares the disparity in the number of heavily burdened planning areas based on core indicators between 30 study sites and the Berlin city level of 542 PLAs. Among them, 36.7% of all investigated refugee accommodation sites have high noise burdens compared to 25.1% of all Berlin PLA levels. It is evident that refugee accommodations are located in areas with above-average noise burdens. Similar results could be found in thermal burdens; 36.7% of all investigated refugee accommodation sites have high thermal burdens, compared to 31.4% of all Berlin PLA levels. These two were also the most common burdens from refugee accommodation sites. In contrast, refugee accommodation sites are less likely to be burdened with air pollution (13.3%), compared to 25.1% for the average PLA of the entirety of Berlin.

**Table 5.** Comparison of the number of highly burdened planning areas according to the core indicators in Berlin.

| Scope | 30 Study Sites | 542 PLAs of Berlin |
|---|---|---|
| **Core Indicators** | **Number of Highly Burdened Sites, Absolute and in %** | **Number of Highly Burdened PLAs, Absolute and in %** |
| Noise burden | 11 (36.7) | 136 (25.1) |
| Air pollution | 4 (13.3) | 136 (25.1) |
| Thermal burden | 11 (36.7) | 170 (31.4) |
| Poor green space supply | 4 (13.3) | 136 (25.2) |
| | 18 (60) Refugee children's perspective | |

As for the green space supply, only 13.3% of all investigated refugee accommodation sites located in areas have poor green space supply, compared to 25.2% of all Berlin PLA levels. It could be indicated from Figure 4 that most of the refugee accommodations are located more on the outskirts with a good supply of green space. However, the number of refugee accommodations with poor access to UGS goes up to 60% when considering refugee children's perspective, i.e., the green spaces are there but not accessible to children because there are barriers like street infrastructure and railways that hinder access.

The most common combinations of sites with twofold burdens are noise and thermal burdens, covering four of the nine sites. *Site 10* has threefold burdens: air pollution, poor green space supply (both perspectives), and social disadvantage. The official departments also identified it as a site needing special attention. *Site 5* has fourfold burdens: air pollution, thermal burden, poor access to UGS, and higher population density (higher degrees of affected inhabitants). Unequal exposure to various environmental burdens and benefits and their uneven distribution among a population with differing levels of vulnerability can heavily contribute to health inequalities. It is particularly pertinent in regions with high cumulative burdens and notable social vulnerability [50]. Therefore, sites have multiple burdens that need to be noticed. More EJB calculation procedures and processes of all 30 study sites can be found in Supplementary Table S4.

### 3.3. Comparison of Findings with Three Study Sites

The neighbourhood environmental levels may provide some detailed empirical materials. *Site 5*, in the city centre, has a high population density ($\geq$20,000 inhabitants/km$^2$, Figure 6a), air pollution, thermal burden, and lower access to UGS from refugee children's perceptions. Also, there is only one attractive UGS from refugee children's perspective with perceived neighbourhood distance for refugee parents. *Site 16* has a medium population density (10,000 to 20,000 inhabitants/km$^2$), twofold burdens, such as thermal burden and social disadvantage, and reasonable access to UGS for refugee children, as there are four attractive UGSs in terms of perceived neighbourhood distance for refugee parents. *Site 25* has a low population density ($\leq$10,000 inhabitants/km$^2$) and no burdens; however, there is no access to UGS from the refugee children's perspective. The authors' previous research has presented evidence that attractive UGS for refugee children is potentially located in higher-population-density residential areas with more road segments [8]; however, in the present paper, these factors may also reflect higher noise burden, air pollution, thermal burden, and limited areas of accessible UGS. Moreover, most investigated sites only have accessible UGS from refugee parents' perceived distances, which means children may only access this UGS when accompanied by their parents [9]. Multi-analysis should be considered for the location choices of refugee accommodation, and those neighbourhoods should be able to provide safe access to UGS from refugee children's perspectives. No further relations between UGS from refugee children's perspective and EJB could be summarised based on current study sites.

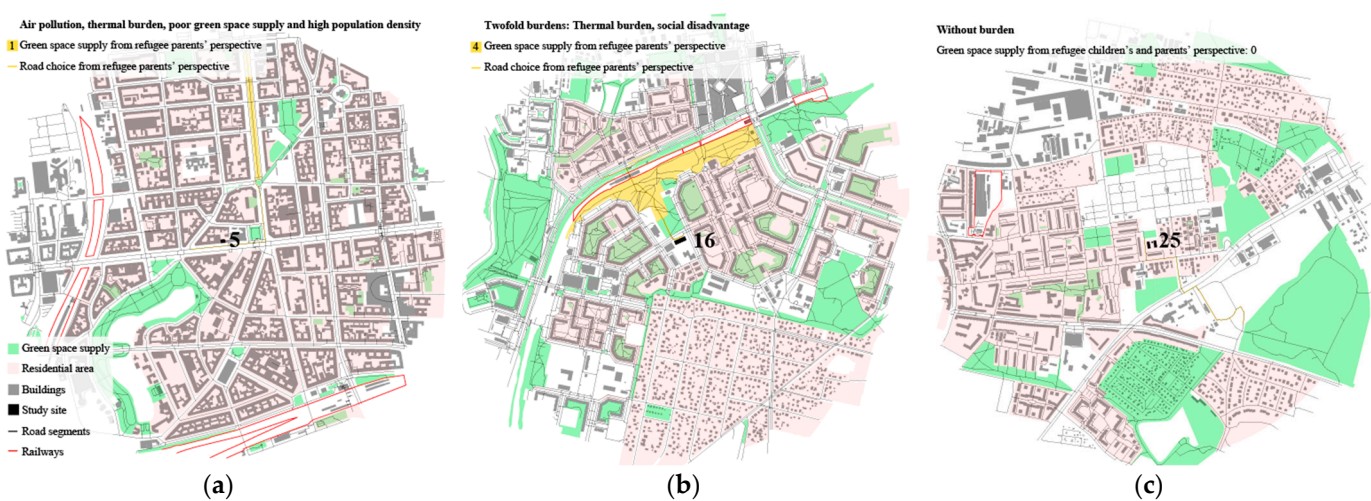

**Figure 6.** Examples of study sites with overall calculation (**a**) fourfold burdens: site 5; (**b**) twofold burdens: site 16; (**c**) without burdens: site 25.

## 4. Discussion

### 4.1. Ensuring the Provision and High Quality of UGS for Refugee Children Who Need It

The World Health Organization [51] suggests that residents in urban areas should have access to 0.5 to 10 ha of public green space within 300 m of their homes. This paper investigated UGS supply conditions of 30 multi-type refugee accommodation sites from refugee children (500 m) and their parents' (500 to 1000 m) perceived neighbourhood distance to UGS of at least 0.5 ha. As mentioned before in the Method section, each Berlin resident should be able to access at least 6 m$^2$ (smaller) to 7 m$^2$ (larger) of green areas. Refugee accommodation locations seem to have better green space supply above average standards (only 13.3% of accommodations have poor access, as compared to 25.2% in all districts). However, the in-depth investigation from refugee children's perspectives shows that 80% of investigated refugee accommodation locations have limited access to UGS. Previous research focused separately on quantitatively developing an environmental justice index involved green space in Berlin [52] or reviewing environmental justice in the context of urban green space characteristics in cities [47]. Kabisch and Haase [21] indicated in their paper that the provision of urban green spaces in Berlin should be centred on a human perspective and sub-districts with high percentages of immigrants have disproportionately lower access to UGS. On this basis, the present research highlights the perceived provision of UGS for refugee children under the environmental justice index. Still, few guidelines focus on guaranteeing the accessibility and usability of UGS for specific socioeconomic and demographic groups, such as refugee children, and this paper has contributed empirical materials to begin such research.

### 4.2. Strengths and Limitations of the Research

Conducting a city-wide analysis with site-specific cases and employing a multi-method approach may necessitate careful and intricate complex consideration. However, the authors posit that the results presented in this paper validate the efficacy of this combination. Methodological enhancements can also be made by incorporating supplementary qualitative and quantitative methods rooted in environmental and social sciences. Quantitative environmental science techniques include the assessment of UGS supply for refugee children, while quantitative social science methods may involve employing conjoint analysis and segmentation methods to summarise refugee children's preferences.

#### 4.2.1. Strengths and Limitations for Size and Representativeness of the Sample

Difficulties existed due to the explorative nature of refugee children settled in refugee accommodations in Berlin. The final data summarisation period was February 2024. How-

ever, overviews and surrounding data from 30 refugee accommodations were collected from 2017 to 2024. Due to the changes in law, temporary transfer, and irregular reconstruction of refugee accommodations, most refugee accommodations had changes in operators, accommodation types, and accepted asylum seeker types. To the best of the authors' knowledge, no official recording was produced to record the number of child asylum seekers in each accommodation. The authors were aware of these difficulties and tried to record the data by obtaining all refugee accommodation lists from Berlin's official department, searching the internet, sending emails, and phoning each accommodation; however, limitations still existed since the authors may have failed to document or include all refugee accommodations which had child residents located in Berlin from 2017 to 2024.

### 4.2.2. Strengths and Limitations for Green space Quality Measures

As mentioned earlier, this paper applied both OpenStreetMap (unofficial) and the Land Use Plan of Berlin (official) to define available UGS in Berlin's research scopes. This design increased the reliability of the data source; however, only two attributes of UGS quality (space size area, the attractiveness of UGS as a destination for refugee children) were evaluated. Still, there was a lack of auditing from other quality key domains, for instance, activities (e.g., type and specific activities for which the space was designated), environmental quality (e.g., the existence of appealing elements), comfort (e.g., the existence of facilities), and physical safety (e.g., indicators and attributes of the adjacent roads). The authors sought to fill this research gap by identifying perceived distance (which related to the social safety of refugee families) in relation to safety [53]; physical safety is still under-researched. Furthermore, the critical domains mentioned above involve subjective perceptions from different social groups and backgrounds; further studies should develop this from subpopulation views of refugee children or other focus groups.

### 4.2.3. Strengths and Limitations for Access to Green Space for Refugee Children Measures

Access to UGS for refugee children was identified with an iterative approach as a supplement to the previous tool the authors developed [8]. The approach was subdivided into UGS availability, accessibility, and attractiveness. It is presented as a quantitative measure index supported by evidence materials mainly focused on perceived safety and simple green space quality, which could be the first step in bringing refugee child-related perceived evidence into academic research; however, there is a lack of further data sources to avail the methodological approach itself, which could be one of the directions for further research.

## 5. Status and Recommendations

Experiencing outdoor space, especially natural space, during childhood and across cultures has beneficial physical and mental health effects [1,54]. As stated by one refugee accommodation manager in an interview, "Refugee children could hardly feel included if they have no place to play". Previous research has identified the poor presence of playspace quantitatively in microenvironments [55], the lack of formal and informal play space in meso-environments [8], and the lack of qualitative refugee children's perceptions [9]. On paper, refugee children have better access to green spaces than the general population; however, in this extended analysis of UGS supply for refugee children of 30 study site neighbourhood environments, we found that refugee children have limited access to green spaces when one considers their specific restrictions in terms of restricted walking radius and safety perception. These selected sites also show their local particularities. Hopefully, the study can offer novel methodological insights into the underlying drivers and causes, as well as offer information for future in-depth analysis or the prioritisation of policy actions in the context of refugee children. The findings from the existing environmental justice system in Berlin can contribute to an environmental equality analysis informing refugee child-related urban policy decisions (for comprehensive refugee accommodation choice and evaluation). Examples of specific recommendations are given below.

*5.1. To Respond to Policy Makers*

The findings show that refugee children in refugee accommodations have unequal resources in urban areas compared to the rest of the population; therefore, the particular aim of creating inclusive UGS should be incorporated into related urban and housing development planning [56]. In *microenvironments*, the authors' previous investigation [55] indicated that the modular refugee accommodations integrate considerations of high-quality UGS into social housing planning for refugee children's play. Moreover, community gardens can provide disadvantaged and vulnerable groups with direct contact with nature and physical activity [24]. Additionally, they provide opportunities for individuals' education, professional development, social integration, and small-scale entrepreneurship [56]. These potential benefits can be demonstrated by the abovementioned Tempohomes offering planting space to every settled asylum seeker [43].

*5.2. To Respond to Urban Planners and Architects*

The findings could aid related participators with the consideration of where and how to employ UGS supplies in *meso-environments*, to include refugee children who live within walking distance of UGS (e.g., where they should build the entrance), and to consider how the resources of UGS are mostly allocated across all potential user populations, including refugee children, for example, playgrounds surrounded by tables and seating, where refugee children could play with supervision, therefore responding to parents' specific safety concerns and providing safe public access to use UGS for refugee children simultaneously [57]. For instance, the abandoned railway may form a barrier to refugee children's access to UGS and may lead to less active play [9]. A positive example is inclusive and comprehensive UGS for all days (e.g., sharing playgrounds and other UGS inside Parisian schools) and service support for vulnerable groups during heatwaves [58].

*5.3. To Respond to Refugee Accommodation and Community Operators*

The social inclusion of vulnerable groups can be supported by participating in the planning of green space; the stakeholders should ensure their expressed needs are taken into account and build their trust in organisational projects [24]. This participation will likely increase future UGS usage [1]. For example, the vulnerable child-oriented project "Les cours Oasis" [59] involved refugee children in co-designing and transforming schoolyard renovation to increase their sense of ownership further.

## 6. Future Research Directions

Through this study, we have identified the status of UGS for refugee children in Berlin. The limited access to the presented UGS and the expected effects resulting from this situation on refugee children should be developed as a form of argument in the future. Despite the limitations of the approach, the authors believe we have demonstrated valuable insights from refugee children's perspectives. Moreover, a suitable connection between research in environmental justice in Berlin and UGS, and the importance of involving specific subpopulations' perspectives for a successful environmental justice approach has been noted. This systematised approach can potentially be used to develop a framework for researching the salutary impact of UGS on diverse populations or merge with other analysis levels, and certain methodological limitations and possible enhancements in future research are emphasised. Future studies could involve more social context indicators (e.g., countries of origin of refugee children). Currently, data restrictions on refugee information remain challenging for the city of Berlin. With the increasing trends of the refugee crisis and their displacements on a global level—and the efforts of host countries for their resettlements—a more comprehensive insight into socio-spatial indicators in additional urban areas would be highly valuable. Only through the comprehensive integration of environmental and socioeconomic information can we achieve a well-founded analysis of urban livelihoods within our cities. Then, it is possible to develop beneficial adaptation measures for vulnerable groups, such as refugee children.

**Supplementary Materials:** The following supporting information can be downloaded at: https://www.mdpi.com/article/10.3390/land13050716/s1, Table S1: Overview of investigated refugee accommodations; Table S2: Data characteristics of category, feature, and coding; Table S3: An iterative calculation of available green space, accessible green space, and attractive green space; Table S4: Procedures and processes of EJB for 30 study sites.

**Author Contributions:** Conceptualization, S.C. and M.K.; methodology, S.C.; software, S.C.; validation, S.C. and M.K.; formal analysis, S.C.; investigation, S.C.; resources, S.C.; data curation, S.C.; writing—original draft preparation, S.C.; writing—review and editing, S.C. and M.K.; visualization, S.C.; supervision, M.K. All authors have read and agreed to the published version of the manuscript.

**Funding:** The APC was funded by ULB Darmstadt.

**Institutional Review Board Statement:** The study was conducted in accordance with the Declaration of Helsinki and approved by the Institutional Ethics Committee of Technical University of Darmstadt (protocol code EK 26/2019 at 9 July 2019).

**Informed Consent Statement:** Informed consent was obtained from all subjects involved in the study.

**Data Availability Statement:** The original contributions presented in the study are included in the article/supplementary material; further inquiries can be directed to the corresponding authors.

**Acknowledgments:** The authors would like to thank Takemi Sugiyama for supervising and all accommodation staff who offered their generous help in this research. The authors would like to thank the reviewers for all their comments and all the editors for all their support.

**Conflicts of Interest:** The authors declare no conflicts of interest.

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
