# Peer review of "Environmental Justice in the Context of Access to Urban Green Spaces for Refugee Children"

_land, doi:10.3390/land13050716_

Round 1

Reviewer 1 Report

Comments and Suggestions for Authors

This is potentially a very important paper, but it needs to be thoroughly revised by a native English speaker - in its current form it is unintelligible, and therefore it is not possible to assess its scientific soundness, merit etc. For example:

Line 44 - Similar forms of natural green spaces around the globe also make them familiar playfields for refugee children from inclusive backgrounds - what do you mean by 'from inclusive backgrounds'?

Line 49 - The policy also recognised the importance of UGS as an environmental contributor for minority groups - which policy? The use of the direct article (The) suggests you are referring to a specific policy. Do you instead mean Policy also recognises ... (i.e. policy in general)?

Line 53 - Providing ‘universal access to safe inclusive and accessible, green and public spaces’ was also considered the Global United Nations Sustainable Development Goal - presumably you mean a SDG, not the SDG. Also, state which SDG

These are just three examples from the first 53 lines of the manuscript ...!

There are incomplete references which are meaningless e.g. 

2. Oxford University Press Green Space. Oxf. Ref. 2017.

35. dpa Notunterkunft Für Geflüchtete in Tegel Geht an Den Start – Berlin.De 2022

And there are issues with the figures too - placing the legend and captions on top of the maps makes them illegible

Comments on the Quality of English Language

This is potentially a very important paper, but it needs to be thoroughly revised by a native English speaker - in its current form it is unintelligible. 

Reviewer 2 Report

Comments and Suggestions for Authors

The text "Environmental justice in the context of access to urban green spaces for refugee children" addresses a very interesting theme, framed within urban planning issues and spatial and environmental justice. The proposed objectives are ambitious, covering different phases of this process, with diagnostic and recommendation components influencing public policies. The proposed methodology is interesting, but its explanation and rationale are unclear. How was the perception of refugees assessed? What is the size and representativeness of the sample? When was the study conducted? What kind of approaches were involved?

This limited methodological clarification, coupled with insufficient conceptual framing - spatial justice, environmental justice, urban planning, multiculturalism - results in somewhat general findings, introducing superficial analyses and responses to the research questions presented.

Therefore, it is recommended that the text be revised, considering:

1. Discussion and conceptual framing of the research, also promoting consideration within the scope of similar studies.

2. Clarification of the methodological approach.

3. Discussion and critical reflection of the results, analyzing, for example, the causes and consequences of the data presented. A comparison with other studies can also be developed.

4. Presentation of recommendations for public policy, considering the identification of problems and resolution scenarios.

Reviewer 3 Report

Comments and Suggestions for Authors

Dear authors

Thank you for submitting your article.

While the themes of spatial justice, UGS and health have become quite common - and necessary - the article addresses and documents research in a particular direction, from the perspective of refugee children, in this case in Berlin.

This is, in my opinion, quite unique. It is a contribution to the topic.

Round 2

Reviewer 2 Report

Comments and Suggestions for Authors

The text has been revised, taking into account the suggestions and comments I have made. The new version is well-structured and demonstrates a notable scientific solidity and coherence. This, coupled with the nature of the research, lends it a high level of scientific interest. I recommend its publication.